# Associations between sleep problems in children with ADHD and parental insomnia and ADHD symptoms

Upasana Bondopandhyay[1], Jane McGrath[2], Andrew N. Coogan[1]*

1 Department of Psychology, Maynooth University, National University of Ireland, Maynooth, Ireland,
2 Dublin and Department of Psychiatry, Linn Dara Child and Adolescent Mental Health Service, Trinity College, Dublin, Ireland

* andrew.coogan@mu.ie

**Data Availability Statement:** Data for this study can be found at https://osf.io/kz3nx/.

**Funding:** The author(s) received no specific funding for this work.

## Abstract

Sleep problems are common in children with attention deficit hyperactivity disorder (ADHD). Children's sleep problem may influence, and be influenced by, parents' sleep problems as well as parents' ADHD symptoms. In the current study we examined the associations of parent-rated sleep quality and sleep timing of pre-adolescent children with parental insomnia symptoms, parental ADHD symptoms and dysfunctional attitudes and beliefs about sleep in a convenience sample recruited by advertisement (N = 120). Childhood sleep problems were common in the sample, with 82% of children exceeding the threshold for the presence of a paediatric sleep disorder. Children's sleep quality showed minimal association with their sleep timing and chronotype. Parental insomnia symptoms, ADHD symptoms and dysfunctional beliefs and attitudes about sleep all associated with their children's sleep quality, and with the sleep subdomains of sleep anxiety and parasomnias. In multiple regression analysis only parental insomnia score was a significant predictor of children's sleep quality. Children's bedtimes, wake times, sleep duration, chronotype or social jetlag did not associate with parents' ADHD or insomnia symptoms. Sleep quality was significantly poorer in children whose parents scored as both consistent for adult ADHD and probable for insomnia disorder compared to parents who scored as either ADHD consistent or insomnia probable, or those who parents scored as neither. We discuss the putative nature of the relationships between sleep quality of children with ADHD and parental ADHD and insomnia symptoms, and suggest that clinicians consider parental sleep when attending to children with ADHD.

## Introduction

Attention deficit hyperactivity disorder (ADHD) is the most common neurodevelopmental disorder in children, with an estimated prevalence of 5–7% in young people under the age of 18 [1]. ADHD is characterised by the core symptoms of attentional difficulties, impulsivity and hyperactivity [2]. Sleep problems are reported to be common in ADHD, with 50–70% of parents reporting that their children with ADHD experience sleep problems [3], including

**Competing interests:** The authors have declared that no competing interests exist.

long sleep-onset latency, delayed sleep phase, increased limb movements, daytime sleepiness, shorter sleep duration and difficulty maintaining sleep [4]. Sleep problems and ADHD symptoms may share common aetiology at multiple levels: dopaminergic dysfunction is implicated in both ADHD and sleep problems [5], and neurocognitive features such as executive dysfunction and inattention, and psychopathological manifestations including internalising and externalising behaviours are reported in both ADHD and sleep disorders [6–9]. These overlapping facets may be the products of complex multifactorial and bidirectional relationships between sleep problems and ADHD symptoms [10,11]. Further, as ADHD is highly heritable [2], and polygenic risk for ADHD and sleep problems overlap [12], there is the possibility that parents and children share overlapping genetic liability for ADHD and sleep problems (but see Lewis et al [13] who do not report increased transmitted genetic liability for insomnia or chronotype).

Sleep problems and ADHD in children have been reported to exert impacts on parental sleep, mental health, and the family context [14]. Sleep dysfunction in children with ADHD may be associated with a decrease in parental wellbeing, and this in turn may decrease parents' ability to implement effective sleep management strategies for the child [15]. Poorer sleep in adults is related to low mood [16], increased anger [17] and increased stress levels [18] which all may impact on parenting behaviours and lead to poorer sleep health in the children [19]. Conversely, children of parents with insomnia are reported to experience higher levels of sleep problems (Zhang et al 2010), as do children whose parents suffer from low mood [20].

As sleep problems are common in adults with ADHD [21], it is of interest to consider the impacts of ADHD-related sleep problems in parents on their children's sleep, and conversely the impact of children with ADHD's sleep problems on their parents' sleep and ADHD symptoms. As ADHD is a highly heritable condition, and there is likely to be a high-level of undiagnosed ADHD in parents of children with ADHD [22], there is a reasonable possibility that parents of children with ADHD will experience greater levels of ADHD symptoms and sleep problems than parents of typically-developing children, independently of issue arising of their children's behavioural challenges.

While previous research has reported that parental sleep quality is associated with their child's sleep and ADHD symptom severity [15,23,24], there are no reports of the relationship between parental sleep and ADHD symptoms, and how these interact and associate differentially with sleep characteristics of their children with ADHD. In the current study we sought to explore whether parental insomnia symptoms, sleep beliefs and ADHD symptoms associate with sleep problems and features in their children with ADHD, with the broad aim of the study being to contribute to the understanding of the family/household context of paediatric sleep complaints in ADHD. We hypothesised that greater parental insomnia and ADHD symptoms associate with greater sleep problems in children with ADHD.

## Materials and methods

### Participants

The sample consisted of 120 parents currently residing in the Republic of Ireland. Inclusion criteria for the study was to be a parent of a child aged between 6 and 12 years old, who had been diagnosed with ADHD by an appropriate clinician. Parents of children with a primary diagnosis of a neurodevelopmental or psychiatric disorder other than ADHD were not included in the study sample. Participants were recruited via purposive sampling through advertising on the online platform of an ADHD support group charity operating in Dublin (ADHD Ireland) between March and June 2021; the advertisement included details that stated that the participants would be contributing to a study on both ADHD and sleep. The study

was granted ethical approval from Maynooth University Research Ethics Committee (SRESC-2021-2429412).

## Study design and measures

The study used a cross sectional design. On receiving written consent for participation, an initial demographic information form was completed by the participant (age and gender of child, age and gender of parent, parent's occupation, whether child is currently on ADHD medication). Following the above, all participants completed a questionnaire on the on-line survey platform Qualtrics, which included a series of validated psychometric instruments detailed below.

**Adult ADHD Self Rating Scale v1.1 (ASRS).** This is an 18 item self-report symptom checklist based on the ADHD DSM-IV criteria, where a subject responds to a particular statement by selecting one of the 5 response options ranging from 'never', 'rarely', 'sometimes', 'often' and 'very often' [25]. Four categories of scores were obtained from the ASRS. The first are two sets of scores for Inattention and Hyperactivity/Impulsivity which are interpreted from the ASRS 18 item scores. These scores are divided into three categories, scores ranging from 0–16 (low ADHD related inattention/hyperactivity), scores ranging from 17–23 (moderate ADHD related inattention/hyperactivity) and scores ranging 24 or above (high ADHD inattention/hyperactivity). A total ASRS score derived from the 18 items of the scale ranging from 0 to 72 was calculated, and an ADHD consistency/inconsistency category was derived. The ASRS is described as having a negative predictive value for clinically-determined diagnosis of adult ADHD of 1 and a positive predictive value of 0.52 [26]; as such, the screener has very strong properties for ruling out the presence of adult ADHD, and considerably more moderate properties for predicting the presence of adult ADHD.

*Dysfunctional Beliefs and Attitudes about Sleep-16 (DBAS)* is a 16-item self-rating scale which was used to assess parents' sleep-related beliefs, as dysfunctional sleep beliefs have been strongly implicated in insomnia disorder [27]. Each item is scored on a 10-point scale; the total score is calculated from all of the items on the scale, with higher scores representing more dysfunctional beliefs about sleep.

*Sleep Condition Indicator (SCI)* was used to assess the presence of insomnia symptoms and the probability of the presence of insomnia disorder in parents. The SCI is an eight-item rating scale developed to screen for insomnia disorder based on DSM-5 criteria, and has been shown to have good psychometric properties [28]. Lower scores on the SCI indicate more insomnia symptoms and a total score of less than 16 indicate probability of insomnia disorder.

**Child Sleep Habits Questionnaire (CSHQ).** This is a 33-item parent report scale to assess the multidimensional sleep problems experienced by children over the past 30 days [29]. Each item in the questionnaire is scored on a 3-point scale as occurring "usually" (i.e., 5–7 times within the past week), "sometimes" (i.e., 2–4 times within the past week), or "rarely" (i.e., never or 1 time within the past week). Sub-scales derived are bedtime resistance, sleep onset-delay, sleep duration, sleep anxiety, night waking, parasomnias, sleep-disordered breathing and daytime sleepiness. A total CSHQ score of over 41 indicates a paediatric sleep disorder; scores of >41 identify 80% of children with a clinically-diagnosed sleep disorder, and the CSHQ is reported to have a specificity of 0.72 for detection of paediatric sleep disorders [29].

*Children's Chronotype Questionnaire* [30] is an adaptation of both the Munich Chrono-Type Questionnaire (MCTQ, [31]) and the Morningness/ Eveningness Scale for Children (MESC, [32]) for use in pre-pubertal children. Caregivers are asked to answer questions about sleep/wake timings on workdays (when the child has to go to school, weekdays) and free days (holidays, weekends, days where no scheduled activities are planned upon waking). Variables

computed are the timing of mid sleep on free days (MSF; which signifies chronotype) and social jetlag (SJL; the difference between the midpoint between sleep onset and sleep offest on free days and work days). This scale also included Morningness and Eveningness (ME) scores that were derived from responses to 10 questions. Morning types were classified by a ME scale score of ≤23, intermediate types by a score of 24–32, and evening types by a score ≥33.

## Data analysis

After screening and cleaning raw data to identify and resolve potential data inconsistencies, scores from each scale were computed and the required total and sub-factor scores were collated in SPSS (IBM Corporation). Descriptive analysis was completed for the data to generate means, standard deviations, percentage frequencies, and composite scores. Data was assessed for normality and presence of outliers using the Shapiro Wilk test and examination of histograms. Legitimate values which were noted as outliers through the box blots were winsorized to 1 x the highest value in that distribution. Spearman's rank-order correlation coefficient was used to determine the relationship between continuous variables. Chi-square tests were used to assess associations between categorical variable. Mann-Whitney U and Kruskal-Wallis tests were used for groupwise comparisons when the dependent variable was not normally distributed, and t-tests and ANOVAs used for parametric data. Missing data was dealt with in a pairwise manner. Multiple linear regression was conducted as a standard model following confirmation of the assumptions of homoscedasticity, normality of distribution of the residuals and absence of high multicollinearity. The target sample size on N = 120 dyads was arrived at following power calculations in G-Power (Faul et al, 2007) on the basis that effect sizes of importance to detect being those of moderate or greater magnitude (eg. Cohen's d of 0.3 or greater for groupwise comparisons). P<0.05 was interpreted as indicating statistically-significant differences and associations, and effect sizes were interpreted as per Cohen (1988). Data for this study can be found at https://osf.io/kz3nx/.

## Results

### Descriptive statistics and children's sleep characteristics

120 parents of children with ADHD were recruited to the study. Table 1 presents the demographic information on the children and parents in the study: 93% of the respondent parents were mothers, and the average parental age was 42.5 years, and 78% of parents were 40 years or older, and the average age of the children was 9.6 years old and ~80% of children were male (average male age was 9.4 years and female mean age was 10.1 years).

   Table 2 shows descriptive statistics for the child and parent psychometric scales: from the CSHQ scores, 82.5% of children were identified as having a paediatric sleep disorder (total CSHQ score >41), and the frequency of such did not differ in different age categories (6–9 years old/10-12 years old; P = 0.79) or by gender of the children (P = 0.322). Chi-square test for independence indicated a significant association between children's ADHD medication use and the presence of a sleep disorder ($\chi2$ = 4.93, P = 0.032, phi = 0.24); 40% of children identified as having a sleep disorder were not on medication, whilst 60% were on ADHD medication (71% of children without a sleep disorder were not on ADHD medication). The Morningness-Eveningness categorisation of children did not differ significantly across age groups, gender, or ADHD medication status (P = 0.271, P = 0.44, P = 0.391 respectively). Female children had a later time of midsleep on free days (MSF) than male children (03:45 vs. 02:45, U = 203, P = 0.009, r = 0.27 (small effect size)), and female children had greater social jetlag than male children (75 minutes vs 25 minutes, U = 219.5, P = 0.01, r = 0.26 (small effect size); these effects of child gender persisted when adding age as a covariate in ANOVA, indicating

**Table 1. Demographics of the children and parents in the sample.** Continuous variables are presented as means and standard deviation, and categorical variables are represented as frequency (percentages).

| Variables | Children | Parents |
|---|---|---|
| Age Mean (SD)<br><br>Age groups n (%) | n-88<br>9.6 (1.96)<br>6–9 40 (45.4%)<br>10–12 48 (54.5%) | n-83<br>42.5 (5.50)<br>29–39 18 (21.6%)<br>40–54 65 (78.3%) |
| Gender n (%) | n-89<br>Male 71 (79.7%)<br>Female 18 (20.2%) | n-101<br>Male 7 (6.9%)<br>Female 94 (93%) |
| Sleep time, Mean (SD) | n-89<br>Male 9:18 pm (1:07)<br>Female 9:30 pm (1:04) | |
| Wake time, Mean (SD) | Male 7:10 am (1:01)<br>Female 7:39 am (0:48) | |
| Sleep duration, Mean (SD) | Male 8.70 hrs (1.16.hrs)<br>Female 8.97 hrs (1.51 hrs) | |
| Parent profession n (%)<br>Medical<br>Non-medical | - | n-84<br>19 (22.6%)<br>65 (77.3%) |
| ADHD Medication Use<br>Yes<br>No | n-86<br>46 (53.4%)<br>40 (46.5%) | - |

that the gender differences are not accounted for by the girls in the sample being somewhat older than the boys).

Additionally, group differences by children's age category were found for MSF (6–9 year old median = 2:30, 10–12 year old median = 3:34, U = 283, P < .01, r = .30 (moderate effect size)) and social jetlag (6–9 year old median = 15 minutes, 10–12 median year old = 60 minutes, U = 384.5, P< .05, r = .20 (small effect size)). CSHQ total score had a moderate negative

**Table 2. Child sleep questionnaire scores and parental scores for insomnia and ADHD symptoms and dysfunctional attitudes and beliefs about sleep.**

| | | Valid N | M (SD) | Minimum | Maximum |
|---|---|---|---|---|---|
| **Child variables** | **CSHQ—Total** | 120 | 53.13(10.46) | 33 | 81 |
| | **CCTQ–SJL (mins)** | 94 | 46.71(41.93) | 0 | 165 |
| | **CCTQ–MSF (hh::mm)** | 91 | 3:02 (1:02) | 1:07 | 5:30 |
| | **CCTQ—ME score** | 96 | 33.34(8.38) | 15 | 47 |
| **Parent variables** | **SCI–Total** | 91 | 19.08(8.55) | 0 | 32 |
| | **DBAS–Total** | 90 | 4.35(1.88) | 0.12 | 8.50 |
| | **ASRS–Total** | 93 | 29.61(16.64) | 0 | 72 |

M- Mean; SD- Standard Deviation; CSHQ- Child Sleep Habits Questionnaire total score; SJL mins- Social Jetlag in minutes, derived from the CCTQ; MSf- clock time of midsleep on free days, derived from the CCTQ; M-E score- Morningness-Eveningness score derived from the CCTQ; SCI score- Sleep Conditions Indicator score; DBAS score- Dysfunctional Beliefs About Sleep score; ASRS–Adult ADHD Self Report Scale total score.

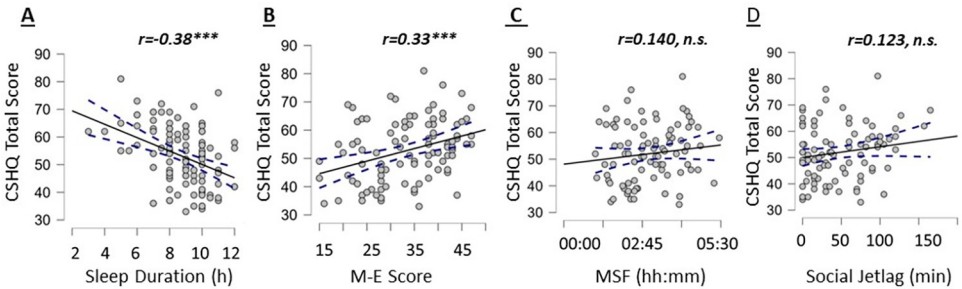

**Fig 1.** Scatterplots showing the associations of children's total CSHQ scores and their (A) sleep duration, (B) M-E score, (C) MSF and (D) Social Jetlag. The filled line represents the regression line, and the dashed line the 95% confidence interval around it.

association with the child's sleep duration (r = -0.38, n = 120, P< 0.001; Fig 1A). A moderate, positive correlation was found between CSHQ and the M-E score (r = 0.33, n = 96, P<0.001; Fig 1B). No statistically significant correlation was found for CSHQ total score with children's MSF (r = 0.140, n = 91, P = 0.186; Fig 1C) or social jetlag (r = 0.123, n = 94, P = 0.238; Fig 1D).

For parents, 36% had probable insomnia based on their SCI scores, and this insomnia probability did not differ significantly as per the parent's age group (29–39 years old vs. 40–54 years old; P = 0.493) or gender (P = 0.496). 35% of parents had scores on the ASRS consistent with the presence of ADHD, and parental ADHD consistency categorisation did not differ as per age group or gender (P = 0.359, P = 0.632 respectively). 22% of parents scored as being consistent with ADHD as well as probable for insomnia disorder, whilst 52% of parents scored as being neither ADHD-consistent or probable for insomnia disorder.

## Associations of children's sleep with parents' insomnia and ADHD symptoms

Associations between children's CSHQ total scores and parental SCI insomnia scores, dysfunctional beliefs and attitudes about sleep (DBAS) and ASRS scores for ADHD symptoms were examined (Table 3). A moderate, negative relationship between children's CSHQ total score and parental SCI scores was found (r = –0.35, P<0.001; Table 3 and Fig 2A). There was also a small, positive association between children's CSHQ scores and parents' ASRS total scores (r = 0.28, P< 0.01; Table 3 and Fig 2B). In addition to the ASRS total score, parental ASRS sub-scores for inattention (r = 0.23, P< 0.05; Table 3), and hyperactivity (r = 0.33, P<0.01; Table 3) were associated with children's CSHQ scores. A small positive relationship was found between CSHQ total score and parent's DBAS scores (r = 0.23, P<0.05; Table 3 and Fig 2C). No significant associations were found between parents' SCI, DBAS and ASRS total scores with the children's bedtime, wake time, sleep duration, social jetlag, MSF and M-E scores (Table 3).

As children's CSHQ total scores were associated with parents' SCI and ASRS scores in bivariate analysis, we further examined the association of the CSHQ subscales with parental scores on the SCI and ASRS (Table 4). Parents' insomnia symptoms correlated statistically significantly with children's bedtime resistance (r = -0.23, P<0.05), as did children's sleep anxiety (r = -0.29, P<0.01), night wakings (r = -0.32, P< 0.01) and parasomnia scores (r = -0.29, P<0.01; Fig 3A). Parents' total scores on the ASRS associated with children's sleep anxiety (r = 0.28, P< 0.01) and parasomnia scores (r = 0.30, P<0.01; Fig 3B).

A linear multiple regression model was run with children's total CSHQ score as the dependent variable and parental total SCI, ASRS and DBAS scores as the independent variables. The

**Table 3. Correlations between child sleep variables and parental ASRS, SCI and DBAS scores.** Values presented as Spearman rank correlation coefficients and their 95% confidence intervals.

| Children | Parental SCI | Parental ASRS | Parental DBAS | Parental ASRS ADHD I | Parental ASRS ADHD H |
|---|---|---|---|---|---|
| CSHQ– Total | -0.353*** (-0.522, -0.157) | 0.289** (0.091, 0.466) | 0.233* (0.026, 0.421) | 0.238* (0.036, 0.421) | 0.332** (0.136, 0.502) |
| CSHQ–Bedtime | -0.051 (-0.256, 0.159) | -0.051 (-0.253, 0.155) | -0.066 (-0.271, 0.146) | -0.026 (-0.229, 0.180) | -0.059 (-0.262, 0.149) |
| CSHQ- Waketime | 0.045 (-0.163, 0.250) | 0.028 (-0.177, 0.230) | -0.079 (-0.283, 0.131) | 0.084 (-0.122, 0.283) | -0.067 (-0.268, 0.140) |
| CSHQ– Sleep Duration | -0.049 (-0.253, 0.160) | 0.056 (-0.149, 0.257) | -0.066 (-0.271, 0.144) | 0.108 (-0.098, 0.305) | -0.014 (-0.218, 0.191) |
| CCTQ– MSF | 0.058 (-0.161, 0.272) | -0.018 (-0.230, 0.196) | -0.168 (-0.373, 0.052) | -0.059 (-0.269, 0.156) | 0.030 (-0.185, 0.243) |
| CCTQ– Social Jetlag | 0.082 (-0.134, 0.290) | 0.042 (-0.169, 0.249) | -0.189 (-0.388, 0.026) | 0.032 (-0.179, 0.240) | 0.089 (-0.124, 0.294) |
| CCTQ– ME score | -0.061 (-0.268, 0.152) | 0.104 (-0.105, 0.305) | -0.017 (-0.228, 0.196) | 0.081 (-0.128, 0.283) | 0.131 (-0.079, 0.331) |

ADHD I–ASRS Inattention item scores total; ADHD H–ASRS Inattention item scores total.

* denotes P< 0.05

** P<0.01

*** P<0.001.

model's adjusted $R^2$ was 0.137 and parental SCI emerged as the only independent variable whose β value was significantly different to zero (β = -0.309, P<0.001; Model 1, Table 5). These results indicate that parental insomnia symptom score is the only significant predictor of children's sleep problems, and that the association between parental ADHD symptoms and children's sleep problems is no longer significant once parental insomnia symptoms are controlled for. Another linear regression model was run, with parental SCI as the dependent variable, and parental DBAS and ASRS scores and children's CSHQ total scores as the predictors; in this model all three independent variables were statistically significant, indicating that the relationship between parental insomnia symptoms and children's sleep problems were at least partially independent of parental ADHD and DBAS (Model 2, Table 5).

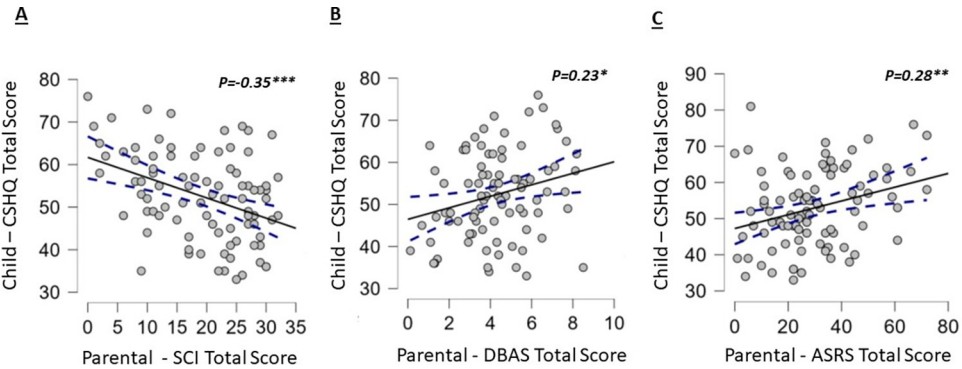

**Fig 2.** Scatterplots showing the associations of children's total CSHQ with parental (A) SCI total scores, (B) DBAS total scores and (C) ASRS total scores. The filled line represents the regression line, and the dashed line the 95% confidence interval around it.

Table 4. Correlations between children's CSHQ sub-scales and parental SCI and ASRS scores.

| Child CSHQ Subscales | Parental SCI | Parental ASRS | |
|---|---|---|---|
| Bedtime Resistance | -0.237* (-0.425, -0.029) | 0.167 (-0.040, 0.361) | |
| Sleep Onset Delay | -0.066 (-0.271, 0.146) | 0.141 (-0.067, 0.337) | |
| Sleep Duration | -0.117 (-0.319, 0.094) | 0.065 (-0.143, 0.267) | |
| Sleep Anxiety | -0.418*** (-0.577, -0.228) | 0.285** (0.084, 0.464) | |
| Night Waking | -0.329** (-0.504, -0.128) | 0.164 (-0.043, 0.358) | |
| Parasomnia | -0.299** (-0.478, -0.095) | 0.307** (0.107, 0.482) | |
| Sleep Disordered Breathing | -0.151 (-0.349, 0.061) | -0.007 (-0.213, 0.199) | |
| Daytime Sleepiness | -0.073 (-0.278, 0.139) | 0.106 (-0.102, 0.305) | |

Values presented as Spearman rank correlation coefficients and their 95% confidence intervals.

* denotes $P < 0.05$

** $P < 0.01$

*** $P < 0.001$.

## Groupwise comparisons based on parental ADHD and insomnia symptom scores

To further examine the relationships between child and parental scores, we undertook group-wise analysis of children's sleep features according to parental grouping based on cut-off points for probable insomnia and ADHD-consistency. Those children with a parent with a ADHD-consistent score did not have higher total score on the CSHQ compared to children whose

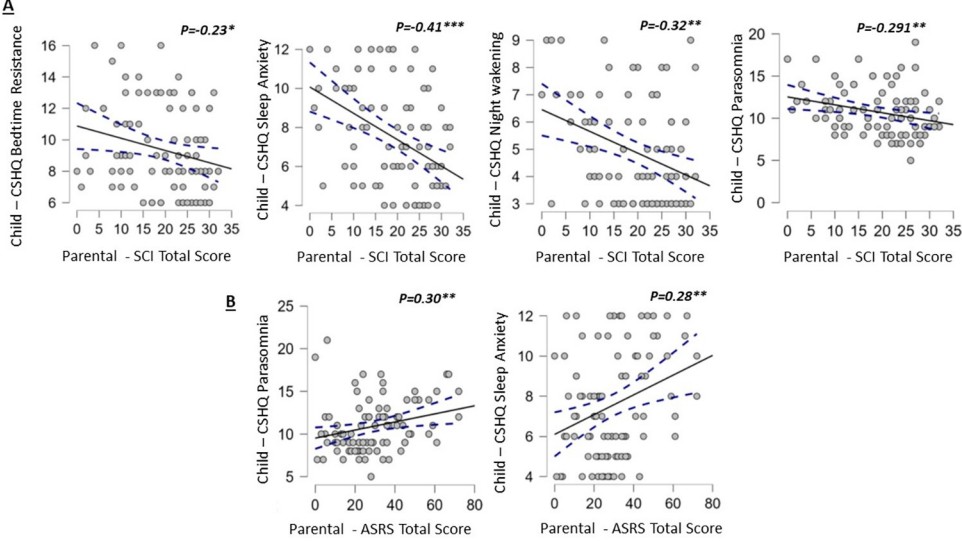

**Fig 3. Scatterplots showing the associations of children's CSHQ subscales scores and parental SCI and ASRS scores.** The filled line represents the regression line, and the dashed line the 95% confidence interval around it.

**Table 5. Model 1: Multiple regression model with total CSHQ score as the dependent variable, and parental SCI, ASRS and DBAS scores as the independent variables; adjusted model $R^2$ = 0.137, F = 5.63, P < 0.001.** Model 2: Multiple regression model with total parental SCI score as the dependent variable, and parental ASRS and DBAS scores and child CSHQ as the independent variables; adjusted model $R^2$ = 0.442, F = 24.21, P < 0.001.

| Model 1: DV = CSHQ | Independent Variable | Beta | t | Sig. |
|---|---|---|---|---|
| | SCI | -.309 | -2.36 | 0.021 |
| *Model R²: 0.137* | ASRS total | 0.112 | 0.388 | 0.348 |
| | DBAS | 0.046 | 0.944 | 0.699 |
| **Model 2: DV = SCI** | **Independent Variable** | **Beta** | **t** | **Sig.** |
| | DBAS | -0.357 | -4.12 | <0.001 |
| *Model R²: 0.442* | ASRS total | -0.354 | -4.03 | <0.001 |
| | CSHQ | 0.199 | -2.36 | 0.02 |

parents did not have an ADHD-consistent ASRS score (consistent n = 32, CSHQ = 55.4±1.8 vs inconsistent, n = 60, CSHQ = 51.1±1.1, P = 0.066; Fig 4A). Children whose parents' SCI scores indicated probable insomnia disorder had higher CSHQ scores than children whose parents did not have probable insomnia disorder (probable insomnia group, n = 32, median CSHQ = 58 vs. not probable insomnia group, n = 58, median CSHQ = 48.50, P<0.001, r = 0.35 (moderate effect size); Fig 4B). Children's sleep duration varied according to parental ADHD-consistency (mean sleep duration of 9.34h±0.31h vs 8.45h±0.21h, P<0.05 respectively; Fig 4C), but did not vary according to parental probable/improbable insomnia grouping (mean sleep duration of 8.74h±0.24 vs 8.72+0.34 respectively, P = 0.95; Fig 4D). Children's M-E scores did not vary significantly according to whether their parents were either ADHD-consistent/inconsistent (P = 0.71; Fig 4D) or probable/improbable insomnia disorder (P = 0.14; Fig 4E).

## Is there an additive effect in the association of parental insomnia and ADHD symptoms with children's sleep problems?

We next examined the association of parental scores indicating both ADHD consistenct and insomnia probability with sleep problem in the children. Children of parents who were both ADHD-consistent and probable for insomnia disorder showed significant worse sleep quality than children whose parents were ADHD-inconsistent and insomnia improbable, and those whose parents were either ADHD-consistent or insomnia probable (F(2, 87) = 8.2, P<0.001; Tukey *post-hoc* test P<0.001 between children of parents who were both ADHD-consistent and probable for insomnia disorder and children whose parents were ADHD-inconsistent and insomnia improbable, P<0.01 between children of parents who were both ADHD-consistent and probable for insomnia disorder and children whose parents were ADHD-consistent or insomnia probable; Fig 5). As such, these analyses suggest that children of parents who scored both as consistent with the presence of adult ADHD and probable for insomnia disorder had worse sleep quality than children whose parents were only one of, or neither of, ADHD consistent/insomnia probable.

## Discussion

The current results show that sleep problems were prevalent in children with ADHD, with 82% of children exceeding the threshold for the presence of a paediatric sleep disorder (a similar finding was recently reported in an independent sample [33]). Sleep problems in children with ADHD were associated with parental symptoms of ADHD and insomnia in bivariate analysis, but parental insomnia symptoms were the only significant predictor of the severity of

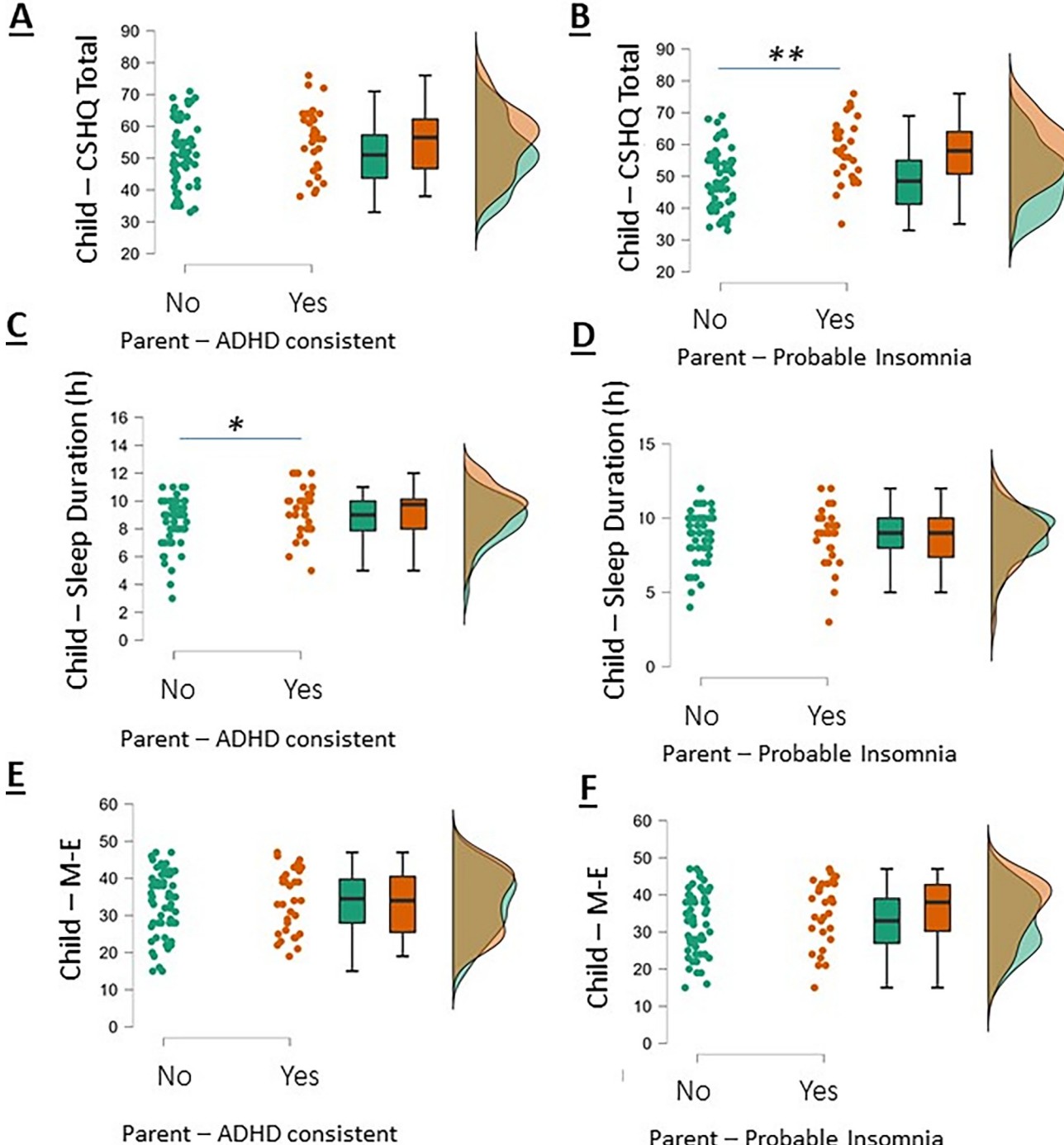

**Fig 4.** Raincloud plots showing groupwise comparisons of children's total CSHQ scores according to parents' ADHD consistency from ASRS scores (A) and insomnia probability from SCI scores (B), children's sleep duration with parental ADHD-consistency (C) and insomnia probability (D), and children's M-E scores with parental ADHD-consistency (E) and insomnia probability (F). ** denotes P<0.01 and * P<0.05 by independent t-test.

children's sleep problems in multiple regression analysis. Children of parents who scored as both consistent for adult ASHD and probable for insomnia disorder had the poorest sleep quality.

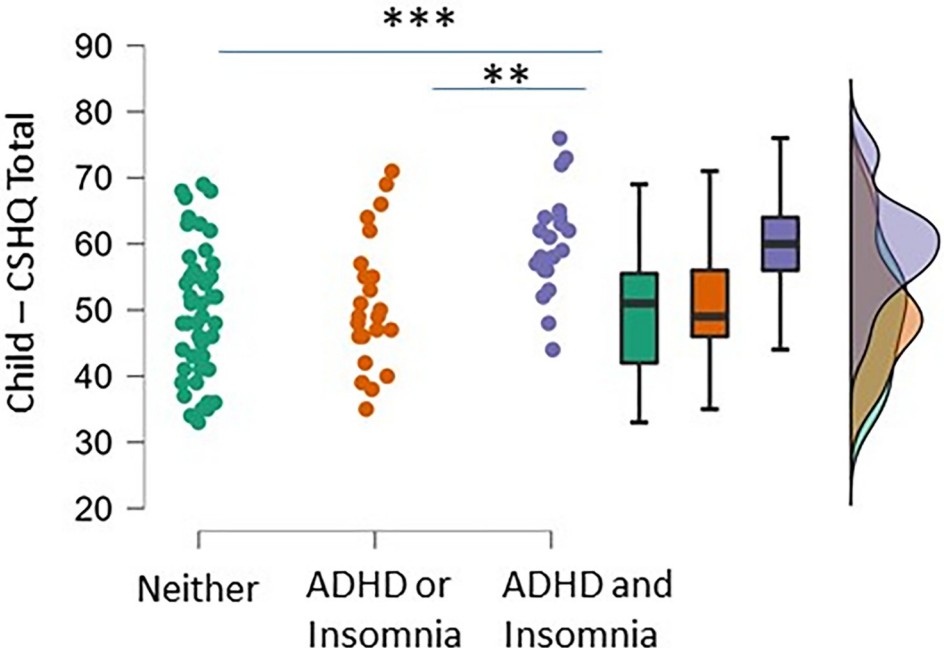

**Fig 5. Raincloud plots showing groupwise comparisons of children's total CSHQ scores according to parents' combined status of ADHD consistency and insomnia probability.** *** denotes P<0.001 and ** P<0.01 by Tukey post-hoc test following one-way ANOVA.

Sleep problems are common in both childhood and adult ADHD [4,21], and sleep problems in ADHD may be influenced by the severity of symptoms, ADHD subtype, comorbid conditions, neurocognitive deficits, socioeconomic circumstance and medication use [34–36]. There is a paucity of literature on the relationship between sleep problems in children with ADHD and parental sleep and ADHD symptoms, although the studies published to date indicate that such relationships may be present and important. Bar and colleagues [37] reported that parental subjective sleep quality was predicted by children's pre-sleep arousal score and anxiety. Parental sleep quality was also found to be associated with total sleep problem scores in a small sample of Japanese children with pervasive developmental disorder or ADHD [38].

Sleep problems in children with ADHD have previously been associated with poorer parental mental health and higher parenting stress [15]. Improvement in parental anxiety is reported to result from behavioural intervention for children's sleep problems associated with ADHD [39]. Lack of consistent daily routines has been reported to predict increased bedtime resistance for children with ADHD [40], and interventions for better sleep hygiene and parenting consistency decreased bedtime resistance [41]. These findings may be consistent with our current report that children's bedtime resistance, night waking, parasomnias and sleep anxiety are significantly correlated with parental insomnia probability, as insomnia disorder is associated with greater sleep timing variability and less consistent sleep routines [42]. As such, greater parental insomnia symptoms may result in less-consistent household bedtime routines which in turn would contribute to children's bedtime resistance, sleep anxiety and night-time wakenings. Further, as Noble et al [40] report that parenting stress predicts children's sleep anxiety, and parenting stress may be associated with parental insomnia [43], parents' insomnia symptoms may contribute to greater parenting stress which in turn contributes to children's sleep problems; a recent study during the early phase of the Covid-19 pandemic suggested a causal link from parental insomnia to children's insomnia [44]. Conversely, children's sleep problems

may contribute to parental insomnia symptoms: children with ADHD are reported to have higher sleep anxiety and more frequent night waking [45], with 22% needing a parent present in the bedroom to go to sleep [46]. As such, children's sleep problems and the resultant demands on parents may contribute to parental sleep problems.

Parents' self-reported ADHD symptoms were also associated with children's sleep problems, although in the regression model only insomnia scores emerged as a significant predictor of children's sleep problems. Insomnia is common in adults with ADHD, with reported prevalence in the range of 43%-80% [47]. Our multiple regression analysis of SCI scores in parents show that children's sleep quality is a predictor of parental insomnia symptoms and independently of parental sleep beliefs and ADHD scores. The current results also indicate that sleep problems were more severe in children whose parent had both probable insomnia and were ADHD-consistent compared to children whose parent was either ADHD-consistent or insomnia probable, suggesting an additive effect of ADHD- and insomnia-symptoms. Features of adult ADHD, such as alterations in time perception, stress and coping strategies may be salient for understanding how parental ADHD may contribute to less optimal children's bedtime routines and sleep habits [48]; however, further work is needed to clarify which particular parental ADHD traits are most associated with children's sleep problems.

Parental expectation of children's sleep duration has been associated with children's total sleep time [49]. Bessey et al. [50] examined parental attitudes and beliefs about their ADHD children's sleep, compared to those held by parents of typically developing children, and reported that parents of children with ADHD endorsed that their child's sleep problems were less modifiable and responsive to change. In our study, we found that that higher levels of parent's dysfunctional sleep related beliefs were associated with them rating more sleep problems for their child. However, such dysfunctional attitudes and beliefs about sleep are common in insomnia disorder [51], and parental DBAS scores did not emerge as a predictor of children's sleep problems independent of insomnia symptoms in the current study; therefore it is not clear from the current results if parental sleep beliefs independently influence children's sleep.

The current study's design means that the directionality of associations between children's sleep problems and parental sleep and ADHD symptoms cannot be ascertained. It seems reasonable to assume that children's sleep problems could impact on parents' sleep routines and quality, and resulting parental sleep problems could increase ADHD-like impairments [11]. Another important point to consider is the potential for shared biological propensity towards ADHD symptoms and sleep problems in parents and children, as ADHD is a highly heritable condition [2]. Recent evidence has suggested that sleep problems in ADHD could emerge due to overlapping genetic predispositions for ADHD and sleep problems [52–55] and ADHD polygenic risk scores were found to be associated with excessive somnolence and difficulty initiating sleep in children [12]. Further, the relationship between polygenic risk for ADHD and ADHD symptoms in children may be moderated by children's sleep duration [56]. However, a recent study reported that children with ADHD do not over-inherit polygenic liability for insomnia or later chronotype [13]; however, as such studies account only for the effects of common genetic variants, the transmitted effects of rare but highly penetrant variants must also be considered. Potential pathways that may link shared genetic liability between sleep problems and ADHD symptoms in both parents and children might include dopaminergic pathways in the pre-frontal cotex, neuroinflammation and iron homeostasis [5]. However, it is worth noting that the variance in sleep problems accounted for by overlapping polygenic risk in ADHD is very low, indicating that other factors (cognitive, behavioural, psychosocial) likely play important roles.

When examining within-child associations, we found an association between CSHQ and M-E scores, but not between CSHQ and MSF or SJL scores; as such, it is not clear from these findings if poorer child sleep in ADHD is associated with greater evening orientation. ADHD, and ADHD symptom severity, in adults is consistently found to be associated with evening preference/later chronotype [21], and later chronotype in adults is associated with poorer sleep quality [57]. However, the current sample was a pre-adolescent one, and given that adolescence is associated with profound changes in chronotype [58], it may be that associations between later chronotype and sleep quality in ADHD emerge only during adolescence. Finally, we found that children's medication status was associated with sleep problems, a finding that has been commonly reported in previous studies [4].

## Strengths and limitations

The current study has some important strengths. We deployed assessments of both paediatric sleep quality and timing in the sample to allow for a robust assessment of sleep issues in children with ADHD. Further, we employed well-validated and clinically-relevant measures of parental sleep problems and ADHD symptoms; combination of measures of ADHD and insomnia in the parents of children with ADHD has not been deployed previously in the examination of links between parental traits and children with ADHD's sleep problems. This is a pertinent issue to address, given both the high prevalence of sleep problems in ADHD and the high heritability of ADHD.

The study also has some important limitations. Firstly, the assessment of children's sleep was solely dependent on parental report through the CSHQ, and future work should seek to include multilevel objective and subjective assessment of children's sleep in the home setting. Further, as the CSHQ is parent-rated, scores may be distorted by parental bias; however, this is a widely used parent-rated scale and is well validated against other sleep measures [59]. Secondly, no clinical information on the severity of children's ADHD symptoms, nor on the subtype of ADHD, was gathered, nor was there information on the type of stimulant medication used (eg. immediate release vs extended release formulations). Further, no information on comorbidities of ADHD in either parents or children were assessed; as such comorbidities are common in ADHD [2] and may impact on sleep characteristics, there is potential for differential findings in the presence or absence of such comorbidities. Socioeconomic status, household composition, family conflict and other potentially relevant social factors were also not assessed in the current study. Thirdly, as the current study design was cross-sectional, causal inferences about the relationships of parental insomnia and ADHD symptoms and children's sleep problems could not be examined, and future work might use longitudinal designs with appropriate statistical modelling approaches to further examined the nature of such relationships. Fourthly, there may be gender-specific features in the relationships between children's sleep and parental ADHD and insomnia features which the current sample was not powered sufficiently to detect. Finally the current study was conducted during the Covid-19 pandemic; given that 34% of parents reported their ADHD children's wellbeing worsening during lockdown, and 31% reported that their children were doing better [60], it is unclear to which extent the current findings may have been influenced by the pandemic, although it is worth noting that schools were open and operating as per usual in Ireland during the period of data collection.

## Conclusion

The current study indicates significant associations between sleep problem severity in children with ADHD and their parents' insomnia and ADHD symptoms. We suggest that clinicians

working with families with children with ADHD may direct some attention to assessing both the children's and parents' sleep, and offer whole-household guidance on promoting healthy sleeping habits.

## Author Contributions

**Conceptualization:** Upasana Bondopandhyay, Jane McGrath, Andrew N. Coogan.

**Data curation:** Upasana Bondopandhyay, Andrew N. Coogan.

**Formal analysis:** Upasana Bondopandhyay, Andrew N. Coogan.

**Investigation:** Upasana Bondopandhyay, Jane McGrath, Andrew N. Coogan.

**Methodology:** Upasana Bondopandhyay, Jane McGrath, Andrew N. Coogan.

**Project administration:** Upasana Bondopandhyay, Jane McGrath, Andrew N. Coogan.

**Supervision:** Jane McGrath, Andrew N. Coogan.

**Visualization:** Andrew N. Coogan.

**Writing – original draft:** Upasana Bondopandhyay, Jane McGrath, Andrew N. Coogan.

**Writing – review & editing:** Upasana Bondopandhyay, Jane McGrath, Andrew N. Coogan.

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
