## [Decision Letter · Decision Letter 0]

23 Oct 2023

PONE-D-23-22588Associations between sleep problems in children with ADHD and parental insomnia and ADHD symptoms.PLOS ONE

Dear Dr. Coogan,

Thank you for submitting your manuscript to PLOS ONE. After careful consideration, we feel that it has merit but does not fully meet PLOS ONE’s publication criteria as it currently stands. Therefore, we invite you to submit a revised version of the manuscript that addresses the points raised during the review process.

 Please submit your revised manuscript by Dec 07 2023 11:59PM. If you will need more time than this to complete your revisions, please reply to this message or contact the journal office at plosone@plos.org. Please include the following items when submitting your revised manuscript:A rebuttal letter that responds to each point raised by the academic editor and reviewer(s). You should upload this letter as a separate file labeled 'Response to Reviewers'.A marked-up copy of your manuscript that highlights changes made to the original version. You should upload this as a separate file labeled 'Revised Manuscript with Track Changes'.An unmarked version of your revised paper without tracked changes. You should upload this as a separate file labeled 'Manuscript'.If applicable, we recommend that you deposit your laboratory protocols in protocols.io to enhance the reproducibility of your results. Protocols.io assigns your protocol its own identifier (DOI) so that it can be cited independently in the future. For instructions see: https://journals.plos.org/plosone/s/submission-guidelines#loc-laboratory-protocols. Additionally, PLOS ONE offers an option for publishing peer-reviewed Lab Protocol articles, which describe protocols hosted on protocols.io. Read more information on sharing protocols at https://plos.org/protocols?utm_medium=editorial-email&utm_source=authorletters&utm_campaign=protocols.

We look forward to receiving your revised manuscript.

Kind regards,

Serena Scarpelli

Academic Editor

PLOS ONE

Journal Requirements:

**Additional Editor Comments:**

Before acceptance some minor revisions are needed.

Reviewers' comments:

Reviewer's Responses to Questions

**Comments to the Author**

1. Is the manuscript technically sound, and do the data support the conclusions?

Reviewer #1: Yes

Reviewer #2: Yes

2. Has the statistical analysis been performed appropriately and rigorously? 

Reviewer #1: Yes

Reviewer #2: Yes

3. Have the authors made all data underlying the findings in their manuscript fully available?

Reviewer #1: Yes

Reviewer #2: Yes

4. Is the manuscript presented in an intelligible fashion and written in standard English?

Reviewer #1: Yes

Reviewer #2: Yes

5. Review Comments to the Author

Reviewer #1: GENERAL COMMENTS

This study aimed to examine the associations of parent-rated sleep quality and sleep timing of pre-adolescent children with parental insomnia symptoms, parental ADHD symptoms and dysfunctional attitudes and beliefs about sleep in 120 parents of children with ADHD.

The article is interesting and is in general well-written.

The strength of the paper is the evaluation of the relationship between parent and children with ADHD with special focus on sleep.

My main suggestion is to emphasize either in the Introduction and in the discussion the importance of the genetic aspects in interpreting the results.

It is well known that ADHD is strictly related to a dopamine dysfunction, and this may be inherited from parents and therefore it is not surprising to find correlation in the same family.

Furthermore, some sleep disorders are associated with dopaminergic dysfunction like restless legs syndrome and periodic limb movement disorders.

RESULTS

This section is very long and some interpretation of the results should be moved in the discussion (i.e first and last paragraph of page 9)

Page 9 line 237-244: this part is confused and not clear please rephrase

DISCUSSION

The discussion is too long and verbose and would benefit of a reduction of some parts.

First line “children with ADHD” please add this

Page 11 line 287 please correct “and”

Page 12 line 311-325. This part is interesting but would benefit of a deeper discussion into the pathophysiological mechanisms that relate sleep and ADHD. It is worth to mention the dopaminergic and the prefrontal cortex dysfunction as the genetic background for the expression of the disorders in both parents and children

Reviewer #2: This manuscript is clearly written and very nicely illustrated with figures as well as tables and arrives at new insights that are supported by data. There is an Interesting discussion about bidirectionality but I miss the role of medication and most so the long-acting stimulants with a high propensity for sleep disturbance.

Abstract

1. Line 25. It would be informative also in the abstract to state that the sample was a convenience sample recruited by advertising

Introduction

2. Line 47 spelling “problems”

3. Line 47-49: the issues are both “long sleep onset latency” and “difficulty initiating sleep”. That sounds like the same kind of thing.

4. Line 74-76: To me, it is more logical to first state that current knowledge/lack of knowledge and then to present what this study intended to address.

Materials and methods

5. Line 88: “psychological disorder” should read “psychiatric disorder”

6. Line 89: Did the ad mention that the focus was sleep problems? (which would increase the proportion of children with sleep difficulties vs a consecutive sample)

7. Line 108: “is” should be “was”. And I don´t get what extra info the next sentence brings. Delete?

8. Line 131 about CSHQ: What about false positives (as was nicely described for ASRS)?

9. Line 132-142: There are abbreviations like M-E score etc. further down and it would be helpful if they are introduced in this section so the reader easily finds them or find them in a list of abbreviations added at the end.

Results

10. 93% were mothers but do you have data on how many was single mothers (who would be expected to be more stressed and experience more sleeping problems)

11. Line 172. Do you have data on what kind of adhd medication and how this relates to sleep issues, i.e. long or intermediate acting or non-stimulants? If so, did long acting stimulants associate with CSHQ? And if associated…to be included in the multiple regression! It is important to separate long acting from short or intermediate acting stimulants for the effects on sleep initiation.

12. Line 174. What was the proportion of meds in the non-sleep disordered group?

13. Line 177: Were the female patients older (as is generally the case)?? Which could explain the later MSF..

14. Line 247: I am just a bit curious regarding the bidirectional associations in this cross sectional study. How about running a linear multiple regression with parental SCI as dependent variable and CSHQ, ASRS and DBAS as independent variables?

Discussion

15. Line 287: spelling “anb d” Your discussion about the bidirectionality could be tested as suggested above (14)

16. Line 301: Spelling: associated actual..

17. Line 357: Another limitation (along with the listed ones) is that the sample contains more boys (4:1) than would be expected (2:1), which might underpower analyses of gender differences.

Table 1

18. It would be of interest to see the age for boys vs girls (as age differences could account for differences in sleep duration, sleep time and wake time.

19. ADHD medication should preferably be split into short/intermediate vs long acting stimulants vs non stimulants

Table 2

20. I suggest a column with the instrument from which subscales are elicited

Table 3

21. Same suggestion as for table 2

6. PLOS authors have the option to publish the peer review history of their article (what does this mean?). If published, this will include your full peer review and any attached files.

Reviewer #1: No

Reviewer #2: **Yes: **Håkan Jarbin

---

## [Author Response · Author response to Decision Letter 0]

8 Dec 2023

Reviewer#1

GENERAL COMMENTS

This study aimed to examine the associations of parent-rated sleep quality and sleep timing of pre-adolescent children with parental insomnia symptoms, parental ADHD symptoms and dysfunctional attitudes and beliefs about sleep in 120 parents of children with ADHD. The article is interesting and is in general well-written. The strength of the paper is the evaluation of the relationship between parent and children with ADHD with special focus on sleep.

My main suggestion is to emphasize either in the Introduction and in the discussion the importance of the genetic aspects in interpreting the results. It is well known that ADHD is strictly related to a dopamine dysfunction, and this may be inherited from parents and therefore it is not surprising to find correlation in the same family. Furthermore, some sleep disorders are associated with dopaminergic dysfunction like restless legs syndrome and periodic limb movement disorders.

Authors’ Response: We agree that shared genetic liability for both sleep problems and ADHD symptoms is an issue of relevance to our current study, which we had addressed in the original version. In line with the reviewer’s suggestion, we have now strengthened sections in both the introduction and discussion (with appropriate nuance to reflect that our study is not a genetic one).

Introduction lines 50-55: “Sleep problems and ADHD symptoms may share common aetiology at multiple levels: dopaminergic dysfunction is implicated in both ADHD and sleep problems (eg. Migueis et al, 2023), and neurocognitive features such as executive dysfunction and inattention, and psychopathological manifestations including internalising and externalising behaviours are reported in both ADHD and sleep disorders (Moreau et al, 2013; Hansen et al, 2013; Accardo et al, 2012; Hansen et al, 2011).”

Introduction Lines 57-61: “Further, as ADHD is highly heritable (Posner et al, 2020), and polygenic risk for ADHD and sleep problems overlap (Ohi et al, 2021), there is the possibility that parents and children share overlapping genetic liability for ADHD and sleep problems (but see Lewis et al, 2023 who do not report increased transmitted genetic liability for insomnia or chronotype).” 

Discussion 340-350: “Further, the relationship between polygenic risk for ADHD and ADHD symptoms in children may be moderated by children’s sleep duration (Morales-Muñoz et al, 2023). However, a recent study reported that children with ADHD do not over-inherit polygenic liability for insomnia or later chronotype (Lewis et al. 2021); however, as such studies account only for the effects of common genetic variants, the transmitted effects of rare but highly penetrant variants must also be considered. Potential pathways that may link shared genetic liability between sleep problems and ADHD symptoms in both parents and children might include dopaminergic pathways in the pre-frontal cortex, neuroinflammation and iron homeostasis (Takahashi et al, 2020). However, it is worth noting that the variance in sleep problems accounted for by overlapping polygenic risk in ADHD is very low, indicating that other factors (cognitive, behavioural, psychosocial) likely play important roles.”

RESULTS

This section is very long and some interpretation of the results should be moved in the discussion (i.e first and last paragraph of page 9).

Author’s Response: We understand this concern, but also feel that it is important to provide sufficient narrative support to the presentation of the results. However, we appreciate that the results section is somewhat long. We have now implemented four subheading to structure the presentation of the results more clearly, and hope this improves the readability of this section.

Page 9 line 237-244: this part is confused and not clear please rephrase

Authors’ Response: We appreciate this comment, and understand it. However, the phrasing used throughout was intended to most accurately reflect the status of the participants in relation to their putative ADHD and insomnia status, and reflects the wording associated with each of the instruments (ASRS refers to symptoms “consistent” with ADHD, the SCI cut-off relates to “probable” insomnia disorder). We appreciate that attempts to use the most accurate phrases leads to a “clunkiness” in places, but we do think this is scientifically justified. 

DISCUSSION

The discussion is too long and verbose and would benefit of a reduction of some parts.

Author’s Response: We have now conducted a further edit of the discussion, and have removed some redundant phrases and passages. Hopefully we now have an appropriate balance of comprehensiveness and conciseness.

First line “children with ADHD” please add this.

Authors’ Response: We have now corrected this (lines 268).

Page 11 line 287 please correct “and”

Authors’ Response: We have now corrected this (line 285).

Page 12 line 311-325. This part is interesting but would benefit of a deeper discussion into the pathophysiological mechanisms that relate sleep and ADHD. It is worth to mention the dopaminergic and the prefrontal cortex dysfunction as the genetic background for the expression of the disorders in both parents and children.

Authors’ Response: As noted above, we have now expanded on these issues in the discussion: lines 340-350 - “Further, the relationship between polygenic risk for ADHD and ADHD symptoms in children may be moderated by children’s sleep duration (Morales-Muñoz et al, 2023). However, a recent study reported that children with ADHD do not over-inherit polygenic liability for insomnia or later chronotype (Lewis et al. 2021); however, as such studies account only for the effects of common genetic variants, the transmitted effects of rare but highly penetrant variants must also be considered. Potential pathways that may link shared genetic liability between sleep problems and ADHD symptoms in both parents and children might include dopaminergic pathways in the pre-frontal cortex, neuroinflammation and iron homeostasis (Takahashi et al, 2020). However, it is worth noting that the variance in sleep problems accounted for by overlapping polygenic risk in ADHD is very low, indicating that other factors (cognitive, behavioural, psychosocial) likely play important roles.”

Reviewer #2.

This manuscript is clearly written and very nicely illustrated with figures as well as tables and arrives at new insights that are supported by data. There is an Interesting discussion about bidirectionality but I miss the role of medication and most so the long-acting stimulants with a high propensity for sleep disturbance.

Abstract

1. Line 25. It would be informative also in the abstract to state that the sample was a convenience sample recruited by advertising

Authors’ Response: We have now added this detail in the abstract: line 24-25 “…in a convenience sample recruited by advertisement (N=120).”

Introduction

2. Line 47 spelling “problems”

Authors’ Response: We have now corrected this (line 48)

3. Line 47-49: the issues are both “long sleep onset latency” and “difficulty initiating sleep”. That sounds like the same kind of thing.

Authors’ Response: Yes, and we have now removed this redundancy (line 49).

4. Line 74-76: To me, it is more logical to first state that current knowledge/lack of knowledge and then to present what this study intended to address.

Authors’ Response: We have re-arranged this section in line with this suggestion. Lines 81-89 “While previous research has reported that parental sleep quality is associated with their child’s sleep and ADHD symptom severity (Martin et al. 2021, Varma et al 2020 Martin et al. 2019), there are no reports of the relationship between parental sleep and ADHD symptoms, and how these interact and associate differentially with sleep characteristics of their children with ADHD. In the current study we sought to explore whether parental insomnia symptoms, sleep beliefs and ADHD symptoms associate with sleep problems and features in their children with ADHD, with the broad aim of the study being to contribute to the understanding of the family/household context of paediatric sleep complaints in ADHD. We hypothesised that greater parental insomnia and ADHD symptoms associate with greater sleep problems in children with ADHD.”

Materials and methods

5. Line 88: “psychological disorder” should read “psychiatric disorder”

Authors’ Response: We have now amended this (line 95).

6. Line 89: Did the ad mention that the focus was sleep problems? (which would increase the proportion of children with sleep difficulties vs a consecutive sample)

Authors’ Response: Yes, and we have no updated the description of the recruitment to reflect this: lines 98-99 “…the advertisement included details that stated that the participants would be contributing to a study on both ADHD and sleep”

7. Line 108: “is” should be “was”. And I don´t get what extra info the next sentence brings. Delete?

Authors’ Response: We have corrected to “was” and removed the redundant sentence (line 116).

8. Line 131 about CSHQ: What about false positives (as was nicely described for ASRS)?

Authors’ Response: We have now included details on the reported specificity of the CSHQ for detecting paediatric sleep disorders: lines 138-139 “..and the CSHQ is reported to have a specificity of 0.72 for the detection of paediatric sleep disorders (Owens et al, 2000).”

9. Line 132-142: There are abbreviations like M-E score etc. further down and it would be helpful if they are introduced in this section so the reader easily finds them or find them in a list of abbreviations added at the end.

Authors’ Response: We have now included the definition of these terms in this section (lines 146-150). Further, these terms are redefined in the legend of Table 2.

Results

10. 93% were mothers but do you have data on how many was single mothers (who would be expected to be more stressed and experience more sleeping problems)

Authors’ Response: We did not collect data on the parental/marital status of the participants, so we cannot address this issue. We do note in the discussion that a limitation of the study is the limited demographic and clinical information that was collected (lines 376-383).

11. Line 172. Do you have data on what kind of adhd medication and how this relates to sleep issues, i.e. long or intermediate acting or non-stimulants? If so, did long acting stimulants associate with CSHQ? And if associated…to be included in the multiple regression! It is important to separate long acting from short or intermediate acting stimulants for the effects on sleep initiation.

Authors’ Response: Unfortunately we did not collect information above and beyond whether the children were on medication for ADHD. This information was self-reported by parents, and we did not have access to clinical records; as such, we did not feel confident in the validity of asking specific details about medication use (dose, formulation, etc). We certainly agree that this is a question of real interest, and there remains limited evidence for the impact of different psychostimulant drugs on sleep (for interest, we have looked at this previously in a sample of adults with ADHD, Coogan et al, Neuropsychopharmacology, 2019).

12. Line 174. What was the proportion of meds in the non-sleep disordered group?

Authors’ Response: We have now included this information: line 183-184 “…(71% of children without a sleep disorder were not on ADHD medication)”

13. Line 177: Were the female patients older (as is generally the case)?? Which could explain the later MSF.

Authors’ Response: We now give the mean ages for boys and girls, and yes girls are somewhat older: line 176 “…(average male age was 9.4 years and female mean age was 10.1 years)”.

Further, we included age as a covariate in the analysis of gender differences in MSF, and controlling for age did not abolish the gender difference, perhaps indicating some other factor such as earlier onset of puberty in girls: lines 188-191 “…these effects of child gender persisted when adding age as a covariate in ANOVA, indicating that the gender differences are not accounted for by the girls in the sample being somewhat older than the boys).”

14. Line 247: I am just a bit curious regarding the bidirectional associations in this cross sectional study. How about running a linear multiple regression with parental SCI as dependent variable and CSHQ, ASRS and DBAS as independent variables?

Authors’ Response: We have now run that analysis with parental SCI as the DV, and parental DBAS and ASRS and children’s CSHQ all are significant predictors, perhaps indicating a relationship between parental insomnia symptoms and children’s sleep quality that is independent of parental ADHD symptoms and sleep beliefs: Table 4 (B) and lines 231-236 “Another linear regression model was run, with parental SCI as the dependent variable, and parental DBAS and ASRS scores and children’s CSHQ total scores as the predictors; in this model all three independent variables were statistically significant, indicating that the relationship between parental insomnia symptoms and children’s sleep problems were at least partially independent of parental ADHD and DBAS (Table 4B).”

Discussion

15. Line 287: spelling “anb d” Your discussion about the bidirectionality could be tested as suggested above (14)

Author’s Response: Corrected, and we have added a section discussing the results of the additional regressions noted above: lines 310-312 “Our multiple regression analysis of SCI scores in parents show that children’s sleep quality is a predictor of parental insomnia symptoms and independently of parental sleep beliefs and ADHD scores.”

16. Line 301: Spelling: associated actual..

Authors’ Response: Corrected (line 320).

17. Line 357: Another limitation (along with the listed ones) is that the sample contains more boys (4:1) than would be expected (2:1), which might underpower analyses of gender differences.

Authors’ Response: We have now included this as a study limitation: line 386-388 “. Fourthly, there may be gender-specific features in the relationships between children’s sleep and parental ADHD and insomnia features which the current sample was not powered sufficiently to detect”

Table 1

18. It would be of interest to see the age for boys vs girls (as age differences could account for differences in sleep duration, sleep time and wake time.

Authors’ Response: Given the small number of girls included in the study, and the point about underpowering noted above, we are not sure that presenting the information in Table one split by gender would be appropriate for this data set.

19. ADHD medication should preferably be split into short/intermediate vs long acting stimulants vs non stimulants

Authors’ Response: As noted previously, unfortunately the only information we have is whether children were on medication for ADHD.

Table 2

20. I suggest a column with the instrument from which subscales are elicited

Table 3

21. Same suggestion as for table 2

Authors’ Response: We have now included in these tables descriptions of which scales the measures were derived from (eg. CCTQ – MSF, CSHQ – Total).

---

## [Decision Letter · Decision Letter 1]

24 Jan 2024

Associations between sleep problems in children with ADHD and parental insomnia and ADHD symptoms.

PONE-D-23-22588R1

Dear Dr. Coogan,

We’re pleased to inform you that your manuscript has been judged scientifically suitable for publication and will be formally accepted for publication once it meets all outstanding technical requirements.

Kind regards,

Serena Scarpelli

Academic Editor

PLOS ONE

Additional Editor Comments (optional):

Reviewers' comments:

Reviewer's Responses to Questions

**Comments to the Author**

1. If the authors have adequately addressed your comments raised in a previous round of review and you feel that this manuscript is now acceptable for publication, you may indicate that here to bypass the “Comments to the Author” section, enter your conflict of interest statement in the “Confidential to Editor” section, and submit your "Accept" recommendation.

Reviewer #2: All comments have been addressed

2. Is the manuscript technically sound, and do the data support the conclusions?

Reviewer #2: Yes

3. Has the statistical analysis been performed appropriately and rigorously? 

Reviewer #2: Yes

4. Have the authors made all data underlying the findings in their manuscript fully available?

Reviewer #2: Yes

5. Is the manuscript presented in an intelligible fashion and written in standard English?

Reviewer #2: Yes

6. Review Comments to the Author

Reviewer #2: (No Response)

7. PLOS authors have the option to publish the peer review history of their article (what does this mean?). If published, this will include your full peer review and any attached files.

Reviewer #2: **Yes: **Håkan Jarbin, MD PhD

---

## [Editor Report · Acceptance letter]

30 Apr 2024

PONE-D-23-22588R1 

PLOS ONE

Dear Dr. Coogan, 

I'm pleased to inform you that your manuscript has been deemed suitable for publication in PLOS ONE. Congratulations! Your manuscript is now being handed over to our production team.

Kind regards, 

on behalf of

Dr. Serena Scarpelli 

Academic Editor

PLOS ONE